# Prurigo Nodularis at Ultra-High-Frequency Ultrasound

**DOI:** 10.3390/diagnostics15131624

**Published:** 2025-06-26

**Authors:** Alessandra Michelucci, Corrado Tagliati, Flavia Manzo Margiotta, Giorgia Salvia, Marco Fogante, Giulio Rizzetto, Antonio Corvino, Elisa Molinelli, Annamaria Offidani, Oriana Simonetti, Marco Romanelli, Valentina Dini

**Affiliations:** 1Department of Dermatology, University of Pisa, Via Roma 67, 56126 Pisa, Italy; alessandra.michelucci@gmail.com (A.M.); manzomargiottaflavia@gmail.com (F.M.M.); giorgia.salvia2@gmail.com (G.S.); m.romanelli@med.unipi.it (M.R.); valentinadini74@gmail.com (V.D.); 2Health Science Interdisciplinary Center, Sant’Anna School of Advanced Studies, Piazza Martiri della Libertà 33, 56127 Pisa, Italy; 3AST Ancona, Ospedale di Comunità Maria Montessori di Chiaravalle, Via Fratelli Rosselli 176, 60033 Chiaravalle, Italy; 4Maternal-Child, Senological, Cardiological Radiology and Outpatient Ultrasound, Department of Radiological Sciences, University Hospital of Marche, Via Conca 71, 60126 Ancona, Italy; marco.fogante89@gmail.com; 5Department of Clinical and Molecular Sciences, Dermatology Clinic, Polytechnic Marche University, Via Conca 71, 60126 Ancona, Italy; grizzetto92@hotmail.com (G.R.); molinelli.elisa@gmail.com (E.M.); annamaria.offidani@ospedaliriuniti.marche.it (A.O.); o.simonetti@staff.univpm.it (O.S.); 6Medical, Movement and Wellbeing Sciences Department, University of Naples “Parthenope”, 80133 Naples, Italy; an.cor@hotmail.it

**Keywords:** prurigo nodularis, dermatology, ultrasound, ultra-high-frequency ultrasound, dermatologic diseases, chronic pruritus, dupilumab

## Abstract

Here, we describe the case of a 48-year-old female patient with prurigo nodularis, where B-mode and color-Doppler ultrasound of one nodule was performed; this revealed hypoechoic dermal and hyperechoic epidermal thickening with lesion hypervascularity. To the best of our knowledge, no previous published articles have reported ultra-high-frequency ultrasound images of this disease, so this case can encourage prurigo nodularis studies in order to better assess ultrasound features and their usefulness in supporting clinical diagnosis and in distinguishing prurigo nodularis from other diseases.

**Figure 1 diagnostics-15-01624-f001:**
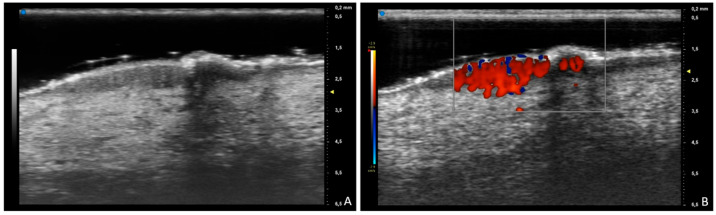
A 48-year-old female patient with chronic pruritus and persistent itching nodules. (**A**) B-mode ultrasound of one nodule shows hypoechoic dermal and hyperechoic epidermal thickening at 70 MHz; (**B**) color-Doppler ultrasound reveals hypervascularity of the lesion. Prurigo nodularis is a chronic inflammatory skin condition characterized by itchy hyperkeratotic crusted or eroded light-to-bright-red papules or nodules from a few millimeters to 2–3 cm in size, with hyperpigmented borders, usually bilaterally and symmetrically located in the dorsal part of the extremities, back and buttocks [1]. The median estimated prevalence of prurigo nodularis is 32.7 cases per 100,000 [2,3]. Dermatological examination is essential to try to identify associated diseases and underlying dermatoses that can be masked by prurigo nodularis. Dermoscopic white starburst pattern surrounding brown-reddish/brown-yellowish crusts, erosions and/or hyperkeratosis/scales can support the clinical diagnosis of prurigo nodularis [4]. Skin biopsy is necessary in prurigo nodularis of unclear origin. The aim of the treatment is to try to interrupt the itch–scratch cycle. First-line therapy for prurigo nodularis includes topical corticosteroids, antihistamines, calcineurin inhibitors and intralesional steroid injection. Second-line treatment is characterized by ultraviolet light therapy and/or systemic treatments such as immunosuppressants, gabapentinoids, antidepressants and mu-opioid receptor antagonists. Adult refractory patients can be treated with dupilumab, which is the first FDA- and EMA-approved drug for this disease [5,6]. To the best of our knowledge, no previous published articles have reported ultra-high-frequency ultrasound images about this disease, so this case can encourage prurigo nodularis studies in order to better assess ultrasound features and their usefulness in supporting clinical diagnosis and in distinguishing prurigo nodularis from other diseases, such as inconspicuous blister in pemphigoid nodularis or the common bullous pemphigoid in early stages, which would show anechoic subepidermal cystic structures with a hypoechoic subjacent upper dermis and no vascularization pattern [7,8].

## Data Availability

Data are contained within the article.

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
