# Peer review of "Prurigo Nodularis at Ultra-High-Frequency Ultrasound"

_diagnostics, 2025, doi:10.3390/diagnostics15131624_

Round 1
Reviewer 1 Report
Comments and Suggestions for Authors
The authors present a clinical case of prurigo nodularis where they describe the sonographic appearence of the prurigo nodules at 47 MHz. Some comments:
- Abstract: Please rewrite the first sentence, as it currently reads that the patient performed US on herself; suggestion "..with prurigo nodularis, where B-mode and color-Doppler ultrasound of one nodule was performed; this revealed...."
- While the authors correctly summarize the current diagnostic possibilities (dermoscopy, skin biopsy), I think it would be relevant to discuss maybe the role of US in differentiating prurigo nodularis from imitators, such as bullous pemphigoid at initial stages/pemphigoid nodularis, which might look similar, having however another pathogenesis and requiring another approach. UHFUS could surely help differentiate between a prurigo nodule and an incospicuous blister in pemphigoid nodularis, which would show anechoic subepidermal cystic structures with a hypoechoic subjacent upper dermis and no vascularisation pattern. (Wang X, Sun J. High-frequency ultrasound features of pemphigoid nodularis: A case report. Skin Res Technol. 2022 Jan;28(1):171-172. doi: 10.1111/srt.13077. Epub 2021 Sep 16. PMID: 34532896; PMCID: PMC9907708.)
- Conclusion: usefullness in supporting the clinical diagnosis is correct, however HFUS is surely also a very good tool which can in vivo, in the right clinical context actually distinguish between prurigo nodularis and pemphigoid nodularis or the common bullous pemphigoid in early stages due to typical sonographic findings. (Porrino‐Bustamante ML, Alfageme F, Suarez L, et al. High‐frequency color doppler sonography of bullous pemphigoid: Correlation with histologic findings. J Ultrasound Med. 2016;35(8):1821‐1825. [)
Other than that, nice article!
Author Response
Dear Reviewer,
Thank you very much for taking the time to review this manuscript. Please find the detailed responses below and the corresponding revisions/corrections highlighted/in track changes in the re-submitted files.
1. Abstract: Please rewrite the first sentence, as it currently reads that the patient performed US on herself; suggestion "..with prurigo nodularis, where B-mode and color-Doppler ultrasound of one nodule was performed; this revealed...."
R: Thank you very much for your suggestion. The first sentence was rewritten as suggested. “with prurigo nodularis, where B mode and color-Doppler ultrasound of one nodule was performed; this revealed hypoechoic dermal and hyperechoic epidermal thickening with lesion hypervascularity”
2. While the authors correctly summarize the current diagnostic possibilities (dermoscopy, skin biopsy), I think it would be relevant to discuss maybe the role of US in differentiating prurigo nodularis from imitators, such as bullous pemphigoid at initial stages/pemphigoid nodularis, which might look similar, having however another pathogenesis and requiring another approach. UHFUS could surely help differentiate between a prurigo nodule and an incospicuous blister in pemphigoid nodularis, which would show anechoic subepidermal cystic structures with a hypoechoic subjacent upper dermis and no vascularisation pattern. (Wang X, Sun J. High-frequency ultrasound features of pemphigoid nodularis: A case report. Skin Res Technol. 2022 Jan;28(1):171-172. doi: 10.1111/srt.13077. Epub 2021 Sep 16. PMID: 34532896; PMCID: PMC9907708.)
R: Thank you very much for this suggestion. A sentence was added: “and in distinguishing prurigo nodularis from other diseases, such as inconspicuous blister in pemphigoid nodularis or the common bullous pemphigoid in early stages, which would show anechoic subepidermal cystic structures with a hypoechoic sub-jacent upper dermis and no vascularization pattern”.
3. Conclusion: usefullness in supporting the clinical diagnosis is correct, however HFUS is surely also a very good tool which can in vivo, in the right clinical context actually distinguish between prurigo nodularis and pemphigoid nodularis or the common bullous pemphigoid in early stages due to typical sonographic findings. (Porrino‐Bustamante ML, Alfageme F, Suarez L, et al. High‐frequency color doppler sonography of bullous pemphigoid: Correlation with histologic findings. J Ultrasound Med. 2016;35(8):1821‐1825. [)
R: Thank you very much for this suggestion. A sentence was added “and in distinguishing prurigo nodularis from other diseases, such as inconspicuous blister in pemphigoid nodularis or the common bullous pemphigoid in early stages, which would show anechoic subepidermal cystic structures with a hypoechoic sub-jacent upper dermis and no vascularization pattern”.
Reviewer 2 Report
Comments and Suggestions for Authors
This short communication presents a case report of prurigo nodularis (PN) evaluated using ultra-high-frequency ultrasound (UHFUS), highlighting imaging characteristics of this underrecognized dermatological condition. The authors appropriately emphasize the absence of previously published UHFUS data in the context of PN, thereby rendering this report both timely and potentially impactful. Below, I outline some concerns:
1. In Figure A, please reposition the identifying letters for each subpanel (e.g., A, B) to appear at the beginning of the corresponding descriptive sentence. For instance: "(B) Color-Doppler ultrasound reveals hypervascularity of the lesion."
2. The authors should clarify in both the abstract and the figure legend the clinical rationale for performing ultrasound in this case, especially considering that UHFUS is not a routine diagnostic modality for PN and there is a lack of supporting literature.
3. The statement regarding the "usefulness in supporting clinical diagnosis" requires elaboration. Specifically, the authors should explain what diagnostic insight was gained through the use of ultrasound. Did the imaging findings help resolve a differential diagnosis, or provide unique information that guided clinical decision-making?
4. The authors are encouraged to comment on whether the UHFUS findings described may be specific to prurigo nodularis, and if so, whether such features could hold potential diagnostic value in future clinical practice.
Author Response
Dear Reviewer,
Thank you very much for taking the time to review this manuscript. Please find the detailed responses below and the corresponding revisions/corrections highlighted/in track changes in the re-submitted files.
This short communication presents a case report of prurigo nodularis (PN) evaluated using ultra-high-frequency ultrasound (UHFUS), highlighting imaging characteristics of this underrecognized dermatological condition. The authors appropriately emphasize the absence of previously published UHFUS data in the context of PN, thereby rendering this report both timely and potentially impactful.
R: Thank you for your evaluation.
1. In Figure A, please reposition the identifying letters for each subpanel (e.g., A, B) to appear at the beginning of the corresponding descriptive sentence. For instance: "(B) Color-Doppler ultrasound reveals hypervascularity of the lesion."
R: Thank you very much for your suggestion. The identifying letters for each subpanel (e.g., A, B) were repositioned to appear at the beginning of the corresponding descriptive sentence.
2. The authors should clarify in both the abstract and the figure legend the clinical rationale for performing ultrasound in this case, especially considering that UHFUS is not a routine diagnostic modality for PN and there is a lack of supporting literature.
R: Thank you very much for this suggestion. In the abstract was added: “and in distinguishing prurigo nodularis from other diseases”. A sentence was added in the figure legend “and in distinguishing prurigo nodularis from other diseases, such as inconspicuous blister in pemphigoid nodularis or the common bullous pemphigoid in early stages, which would show anechoic subepidermal cystic structures with a hypoechoic sub-jacent upper dermis and no vascularization pattern”.
3. The statement regarding the "usefulness in supporting clinical diagnosis" requires elaboration. Specifically, the authors should explain what diagnostic insight was gained through the use of ultrasound. Did the imaging findings help resolve a differential diagnosis, or provide unique information that guided clinical decision-making?
R: Thank you very much for this suggestion. A sentence was added “and in distinguishing prurigo nodularis from other diseases, such as inconspicuous blister in pemphigoid nodularis or the common bullous pemphigoid in early stages, which would show anechoic subepidermal cystic structures with a hypoechoic sub-jacent upper dermis and no vascularization pattern”.
4. The authors are encouraged to comment on whether the UHFUS findings described may be specific to prurigo nodularis, and if so, whether such features could hold potential diagnostic value in future clinical practice.
R: Thank you very much for this suggestion. A sentence was added “and in distinguishing prurigo nodularis from other diseases, such as inconspicuous blister in pemphigoid nodularis or the common bullous pemphigoid in early stages, which would show anechoic subepidermal cystic structures with a hypoechoic sub-jacent upper dermis and no vascularization pattern”.